# Post-Natal Short-Term Home Visiting Programs: An Overview and a Volunteers-Based Program Pilot

**DOI:** 10.3390/ijerph20176650

**Published:** 2023-08-25

**Authors:** Daphna Gross Manos, Noha Gaber Bader, Ayala Cohen

**Affiliations:** 1Social Work Department, Tel Hai Academic College, Qiryat Shemona 1220800, Israel; ayalac@telhai.ac.il; 2Department of Education, Tel Hai Academic College, Qiryat Shemona 1220800, Israel

**Keywords:** post-natal, home visiting, child injury, child maltreatment, child abuse and neglect

## Abstract

Post-natal home visits have been shown to be one of the most effective ways to prevent child maltreatment and reduce risks among children. Unfortunately, these programs tend to be expensive and thus not accessible or practical in many contexts. To address this problem, this paper reviews the literature on home visits conducted shortly after giving birth, considering different types of programs and their outcomes, while focusing on short-term and volunteer-based programs, two approaches that can answer the gap in accessibility. It then introduces a new, innovative, short-term, home visiting program that was developed in Israel. This post-natal program is uniquely structured as volunteer-based to allow it to be culturally informed and inexpensive and therefore accessible to municipalities. The paper describes how experts in the field developed the program and how the volunteers were trained. It elaborates on the protocol for the three defined home visits, each with a specific focus: (1) preventing risks at home, (2) providing parents with emotional support and tools to deal with stress, and (3) connecting them to community resources. We detail the pilot implementation process and some of the challenges that arise. Finally, we describe the design of the evaluation study that is currently collecting data in an Arab town in the north of Israel, with a final discussion on insights gained thus far from the overall process in light of the literature.

## 1. Introduction

The prevention of child maltreatment has become a global health priority as its impact on personal development and functioning is profound, long lasting, and even lifelong, with enormous social and economic costs [1,2]. Early home-visit programs are one of the most effective types of interventions found to reduce child maltreatment rates [3]. However, they are rarely used in a broad and universal way, mainly because of being relatively costly [4,5]. Specifically in Israel, the Social Affairs and Welfare Ministry offers different types of interventions to deal with child abuse and neglect, with a growing focus on prevention [6], but only a small fraction of these services involves a home visit, and these are provided for a very specific high-risk population.

Two possible approaches have the potential to make home visiting intervention more common and accessible, also for systems with limited financial resources. One is to use short-term visiting programs that include only one or two visits by professionals. Growing evidence shows such programs can have an influence on reducing risks among children, lowering rates of investigation for child maltreatment, connecting better to community resources, enhancing parent–child relationships, and improving maternal health [7,8]. The second approach is to involve volunteers to conduct the visits. Volunteers-based interventions to support families and individuals are gradually being studied more, showing some encouraging results on the possible impact, even in place of professional support [9].

Thus, the aim of this paper is to review the current general literature on home visiting programs’ effect in the context of child maltreatment and parental wellbeing, as well as more specifically the literature on the effect of universal programs and short-term post-natal programs. This review also refers to the emerging literature on volunteer-based interventions in general and in the context of post-natal programs. Finally, the paper describes a new pilot program that was developed in Israel based on this knowledge. This short-term program was developed to support every family or parent in a municipality shortly after giving birth through three visits of trained volunteers. It has so far been implemented in one Arab municipality in the north of Israel and a follow-up evaluation study recently started. The process of the program development, its implementation, and the challenges it encounters, are explained together with brief details of the planned research.

## 2. Review of the Literature

The following literature review briefly describes the vast body of established outcomes for early home visiting interventions and then focuses on universal home visiting programs and short-term post-natal programs. Then, it reviews the second approach for making home visiting programs more accessible via volunteer-based home visiting interventions.

### 2.1. Early Home Visiting Interventions

Interventions early in the child’s life offer a rare opportunity to influence the developmental pathway and are found to be effective in reducing reports of child maltreatment, preventing the recurrence of such incidents, and in general reducing risk factors [10]. Home visits usually include individual parenting support in which home visitors implement one-on-one service at homes, especially for women and mothers of young children [11]. Various classical theories, such as human ecology [12,13], self-efficacy [14], and human attachment [15] have influenced the design of home visiting interventions [16].

Home visiting programs have been considered a critical type of prevention program in the context of child maltreatment for some decades in the US. In a review of various child maltreatment prevention interventions, home visitation appears as one of the most effective programs [3]. As far back as 1991, the U.S. Advisory Board on Child Abuse and Neglect recommended that the federal government support a national universal home visiting program for children during the neonatal period, as an effective strategy in preventing child abuse and neglect [17]. A significant budget has been directed in recent years to this type of intervention with the Patient Protection and Affordable Care Act, which established a Maternal, Infant, and Early Childhood Home Visiting Program (MIECHV) that provides a budget for states to establish early home visiting programs [4]. In 2022, USD 337 hundred million in federal funds was spent in 56 US states and territories on home visiting programs that the Department of Health and Human Services considers to be effective [5].

Extensive research has been carried out on home visiting intervention over the years. Meta-analyses and extensive reviews show how programs are effective in reducing child maltreatment, both substantiated and self-reported [3,11], but these results are usually mediated by other factors such as improved parenting skills or better child-parent communication. Evaluation of home visiting programs in different countries found that the programs have positive outcomes on preventing child abuse and neglect by affecting factors such as parental practices and behavior, and improvements in home environment [18,19,20]. Significant improvements were further found in the health and development of young children, including cognition and behavior problems, as well as the reduction of health problems in older children [21]. In one of the most extensive long-term home visiting programs, results showed that the visits improved parental care in different ways. For example, children in the program suffered from fewer injuries and ingestions that may be associated with child abuse and neglect, and they had better emotional and language development. Maternal life also improved after mothers participated in the program. Finally, the program was found to have long-term effects with a reduced number of arrests, convictions, and substance use among the children, 15 years later [16].

### 2.2. Universal Home Visiting Programs

Most home visiting programs have focused on targeted populations with different types of high risks. However, there are some advantages to universal implementation [22]. One of the main reasons for conducting home visits on a universal basis is that targeted programs have not shown consistent impact across sites, and in many cases, success was limited to specific sub-populations with no clear decline in rates of maltreatment for an entire community [23]. Especially as participation rates tend to be limited among targeted programs [24,25], the overall impact is expected to be low, no matter how effective the intervention. Universal programs aim to achieve high penetration rates and population impact, as they have the benefit of a non-stigmatized image and can thus maximize community acceptance [23]. Moreover, as has already been claimed by the US Advisory Board on Child Abuse and Neglect [17] targeted services can promote the withdrawal of public support as the services are not understood as necessary for all.

It should also be noted that the literature is not conclusive in terms of whether home visiting programs are more effective for a population at greater risk. Some suggest such a program is more effective for the at-risk population in specific domains tested [16,26], while others have found the effectiveness of the programs is limited to high-risk children who grow up in disadvantaged families [21].

### 2.3. Short-Term, Universal, Post-Natal Programs

Growing evidence also shows the dramatic effect of universal home visiting programs on a short-term basis. Durham Connects provided one nurse visit covering basic health topics and connecting the family with local services and was evaluated by a large, randomized, controlled trial study. Generally, the program—later named Family Connects—includes a one-time visit by a nurse to new mothers (up to eight weeks postpartum). During the visit, the nurse assesses six different factors: unmet health needs, maternal and infant nutrition, emotional health, substance use, intimate partner violence, and the physical wellbeing of the mother and infant. After identifying the family needs, nurses provide brief intervention, education, and referrals to community resources. Two weeks after the visit, families receive a follow-up call to document the outcomes of referrals made and to assess the need for additional referrals [27,28].

The program had a high penetration rate (80% initiated participation, 69% completed) and fidelity that was empirically assessed using randomized control trials (RCT). Further evaluation after 6 and 12 months showed the intervention group had 50% fewer infant emergency medical care episodes according to hospital records, and fewer hospital overnight stays [23]. The intervention group also had a better connection to community resources, better parent–child relationships, and better maternal health, and blinded observers reported a safer home environment. The program was also found to have a fast cost-benefit impact on the population level through the reduction of infant emergency healthcare costs, saving three dollars for each dollar invested [7,28]. In another evaluation of the program, the primary outcomes revealed families in the intervention group had 44% lower rates of investigation for child maltreatment by child protective services. Regarding maternal mental health, the intervention group’s rate of maternal anxiety or depression was 18.2% vs. 25.9% in the control group [8]. A more recent finding suggests that random assignment to the Durham Connects program had a sustained impact on mothers who self-reported positive parenting behaviors through to the infant age of two years [29].

Long-term effects were also found attributable to Durham Connects. Goodman et al. [30] tried to determine the effect on child maltreatment investigations and emergency medical care when the children were five years old. Their findings show that families assigned to Durham Connects had 39% fewer child maltreatment investigations and 33% fewer total child emergency medical care incidents.

Family Connects programs have been implemented extensively across the US; for example, the Illinois Family Connects [31] and the Welcome Family [27], as well as outside the US [32]. Handler et al. [31] highlight a couple of factors important for the program’s success according to themes that emerged in interviews with new mothers: the value that women, families, and community stakeholders attach to the program, the appeal of its universality, the support for home visiting by nurses in particular, the strategies for engaging women after leaving the hospital, and the initial and ongoing marketing of the program, which may affect women’s willingness to participate.

### 2.4. Volunteer-Based, Home Visiting Interventions

Intervention by volunteers is gradually being reported as clearly successful in various types of support interventions. For example, telephone-based peer support was used to prevent post-natal depression [33]. This involved lay people dealing with depression and anxiety using short problem-solving therapy [34] that was also evaluated using an RCT study [35].

Volunteers’ basic operative programs can have possible advantages, especially in the context of home visits for parents. First, one of the main components of success in home visitation programs is the personal relationship with the visitor [33,36], some even showing that informal support is preferred to that of “experts” [37]. Second, a large part of the knowledge regarding raising children is already known to people in the community from their personal life. Third, programs run by people from the community overcome issues of power inequality between professionals and families and allow families to receive non-judgmental support [36], as well as a better ability to overcome cultural adaptation concerns. Another advantage is the critical financial benefit, as many of the ongoing operating costs can be saved or reduced [17]. Finally, involving people from the community can create positive side effects, such as raising awareness of the issues of possible risks for children in their homes; resourcing knowledge and skills within the community, thus providing a more sustainable solution; and positive outcomes for the volunteers [38,39].

It is important to note that the average effect of programs using health professionals as providers did not differ significantly from those operated by trained paraprofessionals [19]. In another meta-analysis, the qualification of the care provider was not found to be a significant factor in the program’s effectiveness [11].

Supporting these advantages, participants and volunteers in volunteer-based home visiting programs have generally reported high satisfaction with the program and its effectiveness [25,36,40,41]. A study evaluating volunteer home visiting intervention for “vulnerable” mothers in Australia with a small sample found improvements for the intervention group in seven aspects of family functioning, but only two of them were statistically significant: access to social services, and expectation from infants [25]. In Israel, a small survey evaluation of the long-term, volunteer-based, home visiting program, Mom2Mom (M2M), found mothers’ closeness with their volunteers was strongly related to mothers’ gains from the program. They also found the two main reasons for participation were being an immigrant and having a low income, and that the main reported gains from the program were increased self-confidence, improved parenting skills, and communication with the partner [42].

Findings that include RCT evaluation are less common [9], and the few that were carried out show mixed findings. The most extensive evaluation, which involved cluster randomized trials, is that of the long-active Head Start program from the UK [43,44,45,46]. This program offers befriending support to women who are identified before birth as having some risk factors, mainly socio-economic. The evaluation found that the intervention group did show significantly lower levels of parent–child relationship difficulties, but this difference was not accompanied by a change in other expected outcomes, such as enhanced parenting and better use of health services. Some drop was found in levels of stress related to coping with the infant, but not in the general measure of parental stress [44]. Moreover, no differences were found in the child’s cognitive development [43], or in maternal depression [46]. The authors point out that one of the reasons for these findings might be that the intervention was not structured. There are also some limitations of the study design, as program areas and not the families were randomized.

In another RCT evaluation of the Home-Start program, an investigation of a long-term, general friendly support program by volunteers in the Netherlands, a positive change was found in perceived parenting competence, but no effect was found on maternal depression. Parental consistency and observed sensitivity were also found to improve within the intervention group, but no effect was found on other parental measures [47]. In another evaluation of Home-Start in Indonesia, mothers who participated in the program experienced an improvement in maternal emotional functioning [48]. By examining not just the comparable differences between groups, but also the significance of change over time, it was found there was significant change in mothers’ wellbeing and in depression levels. Parental behavior improved significantly according to parents’ reports, although no change was identified in most observational parental measures [49].

Long-term evaluations of Home-Start were also conducted. At the three-year follow-up, the intervention group showed more improvements in responsiveness, affective problem-solving skills of the parents, and less anxiety and emotional problems among the children [50]. At the 10-year follow-up, improvements were found in the feelings of competence, consistency and non-rejection of parenting behavior, and less internalization and externalization problem behavior for mothers in the intervention group [51].

Finally, another older RCT evaluation showed more conclusive results. The evaluation of the Community Mothers Program in Ireland found that seven years after, the intervention group had a lower risk for accidents requiring hospital visits and were more likely to visit the library weekly; parents were not supportive of corporal punishment, felt they could be proud, were more positive about parenthood, and the children were more likely to be immunized [52].

## 3. Literature Summary as Background to a New Program

The findings in the literature show that early home visiting programs are considered effective for reducing risks for children and improving parental wellbeing outcomes. While most of the programs have focused on at-risk target populations, there are some benefits to be considered for providing home visits to the general population on a universal basis. The literature also points out that a short-term post-natal home visiting program can be highly effective in reducing risks for children and promoting wellbeing outcomes for parents. A leading universally based program is Family Connects, which shows long-lasting positive outcomes. Thus, the short-term visiting program shows cost-effective promise. Regarding volunteer-based programs, the literature is less conclusive, and it refers only to long-term programs of “befriending” interventions with no defined structure.

Based on the finding of this review, a pilot program was developed in Israel aimed at combining the different advantages found across the programs in the literature in an innovative way. The developed program was planned to be short-term and structured, open to every post-natal mother in the community, based on a service given by trained volunteers.

### 3.1. New Pilot for Infant Risk Prevention Home Visiting Program Based on Volunteers

The pilot program, named “Citizen Born”, was developed during the year 2019–2020 by researchers from the Child Poverty and Neglect Center in Tel Hai College, in collaboration with the Haruv Institute and in partnership with the “Beterem—Safe Kids Israel” organization. The pilot program was developed to be a universally based, short-term, early home visiting program based on volunteers. The program focuses on establishing contact and offering community support during the sensitive period after delivery, recognizing that for some families this might be a challenging time that can lead to crises.

### 3.2. Main Objectives of the Program

The two main objectives of the program were: (1) to provide postpartum families with basic knowledge regarding babies’ first months of development and prevention of risks; (2) to improve the relationship of the municipal authority with postpartum families, strengthening parental contact with services in the community according to their needs.

### 3.3. Intervention Description

During the first three weeks after delivery, families are contacted regarding their participation in the program. Families who agree to participate receive three structured home visits from the volunteers. During the meetings, the volunteer does a short basic assessment of the family’s needs, and emphasizes safety, child development milestones, relationship with the baby, and available local resources for post-natal families. According to the most pressing needs suggested by the family, the volunteer connects them to relevant local services. Two weeks after the third home visit, the volunteer makes a phone call to ensure that the family managed to engage the services in which they were interested, and, if necessary, makes direct contact with the service.

### 3.4. Main Principles of the Program

The program is based on the principle of respectful, nonjudgmental interaction with families, while emphasizing cultural sensitivity and adaptation. The message conveyed is that parenting is not an easy job for anyone, giving a sense of legitimacy to difficulties and a feeling that they are not alone.

### 3.5. Content of the Visits

A full description is in Appendix A.

First visit: personal introduction, awareness of the condition of the baby and the parents, identification of needs, and initial familiarization with services. Main content: (1) Assessing the basic situation of the family after delivery, recognizing possible challenges. (2) Raising awareness of the importance of paying attention to the baby’s mood and temper. (3) Raising awareness of the difficulties that may arise in the face of the baby’s crying and suggesting possible ways to cope with it. (4) Identifying general needs after giving birth, including economic. (5) Checking the family’s familiarity with relevant services in the municipality and referring them to relevant specific services if needed.

Second visit: a short visit based on a slightly longer program developed by “Beterem”, focusing on home safety for children. Main content: (1) Identifying parents’ attitudes towards child safety and accident prevention. (2) Empowering and building supervision capabilities. (3) Engaging parents in child safety acts.

Third visit: the importance of the parents’ interaction with the baby and the mental state of the parent. Main content: (1) Acknowledging the importance of interaction with the baby and emphasizing how the parent can be more engaged in such activities. (2) Raising awareness of the impact of a parent’s mental state on the baby, and how important it is to seek help if needed. (3) Awareness of the change in the couple’s post-partum relationship and the need for adjustments. (4) Program summary and feedback.

### 3.6. The Program Development Process

The starting point for the development of the program was that we researchers recognized the accumulating evidence in the literature, as shown above, for the impact home visitation programs have on reducing risks among children, even short programs such as Family Connects. In Israel, there is a basic health promotion service, given to newborns and infants by nurses, not based on home visits but rather relying on the parents’ scheduling visits in the nurses’ clinic. When it comes to professional home visits, very few local programs exist, and only on a sporadic basis, mostly intended for families identified as a “high risk” by the welfare agencies. By conducting meetings with relevant agencies in the field we soon realized that home visits by nurses are not relevant to the Israeli context, mainly because there is a very good universal service provided by the health system in which families receive health support in the first two years during family visits to the local clinic. Furthermore, the necessary budget for such a program is very high and irrelevant in the Israeli context, which has seen a decrease in social governmental expenditure in the last two decades [53]. At the same time, we noticed growing evidence for the advantages of programs based on volunteers, especially when resources are limited, in better community building, providing services by people who are more approachable, and sometimes in improving recruitment rates as there is less stigma than when welfare professionals conduct the visit.

*Needs survey*: To better understand the possibility of such a direction for intervention and to better understand the family’s needs, in the specific context of families in the north of Israel, we conducted a preliminary basic needs mapping survey in Kiryat Shemona, a small town. This survey included 52 participants and identified families’ needs and interest in that kind of service. They thought that entry around the age of one month is relevant, preferably by a female volunteer, and were supportive of the idea that people and even they themselves would be ready to be involved in such program.

*Developing visits’ content*: The next step was to develop the protocols of the visits, based on former research and a discussion with a team of experts, mainly from Haruv Institute, to develop the relevant contents of the program and training. This group included a social work expert with a PhD in child–parent psychotherapy, a clinical psychologist expert in mentalization-based treatment, a developmental psychologist, a public health promotion nurse, a parent support group instructor, and a social worker expert in parent PTSD. A series of meetings with “Beterem” was conducted to incorporate their contribution, mainly during the second visit. “Beterem” has vast experience with home visitation programs in the context of general risks for children at home. Moreover, including a general home safety part in the program would contribute to “normalizing” the program and provide broader coverage of the issue of risks for children at home. After a long process, the program content was finalized in terms of length and depth of content, which is practical for short-term programs.

The next step was trying to implement the program in relevant towns in the north of Israel. We reached out to three municipalities to conduct this pilot: a secular Jewish city, a mixed ultra-Orthodox Jewish city, and an Arab city. In a series of meetings with the municipality’s early childhood center and mayors, the program was introduced to them, and they were asked to support it by providing a part-time (25%) coordinator for the program, to recruit suitable volunteers for the program, train them, and later coordinate the program implementation.

Aspiring to emphasize universality, we aimed to offer the program to all families from these three municipalities who give birth in the two main hospitals in this northern area. We further planned to conduct evaluation research (its structure will be described later) and thus reached and obtained the approval of the Helsinki Committees of the two hospitals. As part of this process, we developed the study and program recruitment form as well as the study questionnaire in Hebrew and in Arabic.

Unfortunately, at the same time the ethical process was carried out, we found out that one municipality could not come up with the budget to support a coordinator for the program. A couple of months later, the second municipality dropped out two weeks before training, as that was when COVID-19 started, and they decided not to move the program online as they did not think their volunteers would be able to manage it. So eventually only one Arab municipality started the program with 20 female volunteers participating in online training supported by Haruv Institute. Their ages varied between 35 to 65. Twelve are Druze, six are Muslim and two are Christian. Ten of them had an academic education (three social workers and seven teachers), eight had other types of higher education, and two had a high school education.

*Volunteer training*: The training included six meetings. The first meeting focused on introducing the program and its main goals and listening to a lecture by a developmental psychologist on the importance of the first year and family needs after delivery. The second and third sessions focused on supporting a safe environment at home, given by a specialized trainer from “Beterem”. The fourth session included a pediatrician from the child emergency health center from one of the main hospitals in the region, on signs of child abuse and neglect, and a lecture on families’ mental health in the first year by the developmental psychologist. The fifth session focused on home entry and communication skills needed during a visit by an early childhood expert. The last session included representatives of different relevant services for families with babies in the municipality: well-baby clinics, child development services, early childhood centers, and the welfare department and health department. At the end of the training, there was a graduation ceremony in the presence of the mayor and a gift from the municipality.

### 3.7. Program Implementation Process and Challenges

The end of the training marked the beginning of the implementation phase, and the attempt to recruit families from the Arab municipality who gave birth in the relevant hospital. At the hospital during the discharge process, the department secretary reached out to mothers and their partners asking them to participate in the study and possibly also the program, with a promise of some small compensation at the end of the research, and an Arabic-speaking nurse available for more explanation. However, after about two months when no family was recruited, we realized we were experiencing another major challenge. One main reason seems to be that the number of births from this municipality significantly dropped during these months, as we understood most of the women from this town were now giving birth in another smaller hospital in the region. After another month we decided to stop recruitment through the hospital and approach our college ethics committee to recruit families through community services.

Recruiting through community services such as community health care clinics, the welfare department, education services, and municipality social networks worked much better, and finally, the program could start. However, unfortunately, the coordinator role was canceled under the reduction of the state budget for the Arab sector, due to a change of government. A new temporary coordinator for the welfare office was assigned to the role, but in a very partial role (about 10%) so the progress was slow. As the volunteers started to conduct the visits, we understood it was necessary to adjust the volunteer’s religion to match the family’s religion (Druze/Christian/Muslim). Furthermore, a request came from the volunteers to enter homes in pairs, saying they felt more competent and comfortable this way, and better able to support the mother.

### 3.8. Planned Study and Evaluation

We recently completed the pilot phase and have begun to follow up the program with an evaluation study. Families from the city who gave birth during the preceding month were approached by different services in the city and invited to participate in the study. They were told that the study includes questionnaires for one of the parents to answer at two different points in time, and that the study also includes three home visits by volunteers. Families who gave their consent were randomly assigned to the control group (one-third) and study group (two-thirds), and every third family is in the control group. Both groups are supposed to fill in the first questionnaire when the baby reaches the age of one month. Then the visits should be conducted for the study group between the ages of one to four months. The control group will fill out the second questionnaire when the baby is between three and four months old. The study group will be approached for the initial home visit and follow-up home visits as needed. Upon completing the second questionnaire the families will receive a small gift card (NIS 50). We aim to reach a sample of 100 babies in total (The study process is depicted in Figure 1).

Evaluation measures: The quantitative part includes a questionnaire before and after the program focusing on the evaluation of three main levels of outcomes: community contact and services, parents’ wellbeing, and child risk perspectives. *Community:* Satisfaction from neighborhood services (Neighborhood Factors and Child Maltreatment Study questionnaire [54], Community Connectedness Scale [55], and Collective Efficacy Scale [56]. *Parents:* Personal Wellbeing Index [57], Parental Stress Scale [58], and Parenting Sense of Competence Scale [59]. *Child:* Home Safety Questionnaire [60]. We also add a general questionnaire for the program evaluation from the Mom to Mom program [61].

The qualitative evaluation will incorporate multiple perspectives regarding the program. It aims to gain a holistic understanding by examining perceptions, worldviews, and meanings that are influenced by the participants’ subjective interpretations [62]. As part of the qualitative analysis, 14 interviews will be conducted with participants in the program asking about their perception with regard to the program’s content, relationships with the volunteers, issues requiring improvement, and assistance that was valued. The interviews will be audiotaped, fully transcribed verbatim, and anonymized. The texts will be read and re-read several times. They will be coded according to recurring themes, which will be mapped according to methodically identified interconnections and emerging patterns [63].

## 4. Discussion: Raising Issues in Program Implementation

The development and implementation of this new program raise some issues that should be considered when trying to engage in similar programs, especially in unique cultural settings. We will discuss some of these main issues that uncover some of the main possible strengths of the program as well as possible emerging weaknesses that need to be addressed.

### 4.1. Cultural Adaptation

A couple of complexities arose concerning cultural adaptation. First, as the first site that agreed to participate in the program (even as COVID-19 started) was an Arab town, the materials had to be translated from Hebrew into Arabic followed by a close scrutiny from the local team for any necessary adjustments. Furthermore, the training was conducted in Arabic and with local Arab professionals. However, besides the more technical issues, two further cultural challenges emerged as we progressed with the program. First, as mentioned, we were attempting to provide a universal program, so we planned to recruit families through the hospital, but unfortunately it seemed that families from the town approached by the staff at the hospital were not willing to participate. Of course, this might have been an issue of timing due to COVID-19 uncertainty, but as other programs had successfully approached families at the hospital a short time after giving birth [22], we suspect there might be some cultural intricacies, perhaps related to the complex relationships between minority populations (the Arab families giving birth) and majority populations (the mainly Jewish hospital staff) [64]. The second challenge arose when matching volunteers with families: the volunteers insisted that each volunteer and family be of the same religion. As the chosen town boasts three different religions—Druze, Muslim, and Christian—this added complications to the matching process.

### 4.2. Dealing with Unexpected Events

The development process of the program was thorough, involving a literature review, the involvement of an interdisciplinary team of experts, and many meetings with the relevant early childhood centers in the field. However, we were unable to predict two unexpected events that had a major effect on the implementation of the program. One was COVID-19, which unfortunately led to the second municipality cancelling their participation in the pilot. The second unexpected change was that by the time we started the program, the main hospital where the women from the town went to give birth changed (as a well-regarded doctor from the town started to work in the relevant department in another smaller hospital), and was no longer the one with which we had made contact and for whom we received Helsinki approval (which we chose according to the birth statistics). It is important to take into account that as much as you try to plan ahead there are always changes at the macro and micro levels.

### 4.3. Reciprocal Relationship with the Community

A major support was the close connection of one of the authors to officials in the municipality. This especially helped when the program moved from hospital-based recruitment to community-based recruitment, in introducing the program to all the relevant local organizations and reaching out to relevant families. Her involvement helped to keep the early childhood center engaged during the disruptions due to COVID-19, and many other small implementation challenges. We were surprised by who volunteered, having originally thought the dominant profile of volunteers would be retired seniors, not necessarily with a strong educational background. However, most were educated working women, coming from social and educational fields, who felt the program was important. In addition, we were further surprised by the contribution of the program to the volunteers themselves. This reciprocal relationship with the community can be enhanced in the future by a steering committee that can support the program’s implementation and the ongoing connection with the local stakeholders.

### 4.4. Program Resources and Affiliation

When we approached the municipalities, we asked them to support the program with a 25% coordination position. While they agreed—and at the beginning, there was a budget for such a coordinator within the early childhood center—after a couple of months the Israeli government changed and that budget was no longer available to the municipality. Then the municipality supported the program with a coordinator who was a social worker in the welfare department, and that brought a lot of complications, as naturally, she was very busy with other aspects of her work. Only lately another arrangement for a coordinator was set, once again within the early childhood center. Thus, it seems that having a more stable designated budget for the program is critical, and while the close connection with the municipality is very important for its implementation, it might be better to have a reliable and independent budget coming from charitable foundations or federal outlay. We also found it important for the image of the program to be less affiliated with the municipal welfare department.

### 4.5. Adherence to Study Protocols

As the program was developed by an academically orientated team, the implementation process was intended to incorporate an evaluation study from the very first step. As the interest and priority in the field are naturally different, we encountered issues regarding the implementation of the study, some of which are still being addressed. First, for the program community team and the volunteers, who were very enthusiastic about the program, it was not easy to have a control group, as they had to deliberately not serve families in their own community. To deal with this, we had to explain the purpose of having a control group for the local team and the possible long-time benefits. We also allowed them to offer the program to families in the control group who finished the second questionnaire whose child was younger than four months old (the program content is not relevant for older babies). Furthermore, the need to separate the study team from the program team was complicated for many of the volunteers to understand, as they thought it does not make sense to have another person (from the study team) involved with the family in this complicated time after birth, especially considering cultural and religious sensitivities. Again, we had to explain the rationale, taking into account the complexity of implementing the program in a small, highly interconnected community, where the people all know each other. Finally, another ongoing challenge is fidelity to the program’s protocol by the different volunteers. As more time passes since the training took place, this is becoming more difficult. We currently deal with it by supplying continuous messages emphasizing the importance of following the protocol accompanied by support meetings and follow-up training for the volunteers.

## 5. Limitations

While the program tried to address some of the accessibility problems of home visiting programs and manifest its vast strengths, the current program has some limitations. First, at this point, it was only implemented in one town, in a specific context: Arabs, particularly of Druze ethnicity. Implementation in another place would have brought up other issues. The current program still does not have strong budgetary support for its implementation, which raised more complexities in the process. Finally, the study is only in a very early stage, so we are still awaiting the possible long-term effects of the program as described by the literature, and we do not yet know its actual impact.

## 6. Conclusions and Recommendations

The current manuscript reviewed significant evidence in the literature for the impact a home visiting program can have on preventing risk to children and promoting parents’ wellbeing. It further brought more specific evidence that shows the possible impact short-term programs and volunteer-based programs can have, two possible approaches that can provide a budget-appropriate implementation of post-natal structured home visiting programs. It then presented a program developed intentionally to cover both aspects. While more implementation sites are needed, as well as the longer-term results of the evaluation study, we can already draw some important conclusions regarding the implementation of similar programs based on the process we detailed here.

During the program implementation, we found that close attention to cultural aspects is critical, especially in the complex Israeli–Arab and multi-religious contexts. Implementation of a program within a community also demands flexibility to micro and macro changes and developing sustainable close connections with the community. Attending to budgetary issues was also found to be crucial, even for a program that from the outset was developed to be cheap to run.

Generally, in terms of the initial recommendations for the implementation of similar programs, we found it valuable to have a close relationship with the community to ensure cultural adaptation and make adjustments when necessary. We also found it important—while trying to keep a certain structure in the program—to attend to the volunteers’ important suggestions, as they may determine the program’s success in practice. As part of this, making sure volunteers have ongoing support and connection with one another also seems to be critical. Finally, we found it is important to establish strong support for the program by different stakeholders both within and outside the community.

## Figures and Tables

**Figure 1 ijerph-20-06650-f001:**
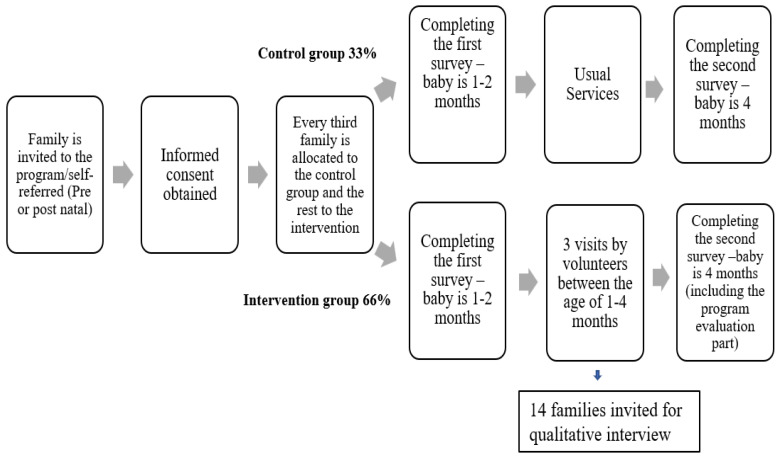
Planned study participants’ flow diagram.

## Data Availability

Not Applicable.

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
