# Peer review of "Post-Natal Short-Term Home Visiting Programs: An Overview and a Volunteers-Based Program Pilot"

_ijerph, 2023, doi:10.3390/ijerph20176650_

Round 1

Reviewer 1 Report

On the whole, a thought provoking, interesting and important article which has wider  relevance.

The title of the article too long, I would suggest shortening it. 

Author Response

We thank the reviewer for his encouraging comment. The title was revised and shortened to:

Post-Natal, Short-Term, Home Visiting Programs: An Overview and a Volunteer-Based Program Pilot

Reviewer 2 Report

Thank you for the opportunity to review the article: Post-Natal Short-Term Home Visiting Programs: An Overview and a New Pilot for a Risk Prevention Program Based on Volunteers. The topic is very interesting and valuable; however, I think the article should be written once again with many changes.

In this form, it is an elementary review of the literature and an announcement or only a description of the actions taken in a planned program of postnatal home visits. We can read anything about the results; we can not see any statistical analysis, effects, or conclusions of this research. It is a first step and may not be the best time to publish. Moreover, the article is very long and, in large part, is a description of the struggles of researchers in the implementation of specific research goals.

Perhaps, the suggestion to create two independent articles will be a hint. One with a good literature review and the other with a description of your program and its effects when you will know the results. The thought of comparing different cultures seems very interesting, but the results of the study only from Arabic city is also a good idea. Good luck!

Author Response

We thank the reviewer for the comment. We are, of course, aware that the paper does not include any results at this point. However, we have decided to answer the call for papers for this special issue and to focus on the literature review and preliminary description of the program as we believe it is informative. As scholars who are also involved in implementing programs in the field, we know that descriptions of challenges encountered and possible solutions can be very useful. We do plan, as recommended by the reviewer, to dedicate a separate paper to the program evaluation (and to make things clearer, we omitted the one paragraph that started to discuss some of its effects in this paper – see the answer to Reviewer #3).

Reviewer 3 Report

The authors have carried out a good research work in which they combine a good literature review with a pilot home intervention program, carried out by trained volunteers. I congratulate you.

In order to improve the comprehension of the manuscript, I make the following suggestions:

1. I think it would be convenient to change the title of section: 3. Literature review summary, for another title that better fits the content of the section.
2. Although in section 4.3 it is mentioned, it would be important to explain in greater depth the characteristics of the volunteers (training, age, etc.).
3. As the program is at an initial stage, it has not been possible to evaluate the long-term effects, but neither have short-term effects been observed, as was the initial objective? Two concrete examples are given in point 5.1, but it would be useful to present the results of the intervention in a more structured and concrete way.

As positive points of the work it is necessary to indicate the following:

- The title does accurately reflect the content of the article.

- The intervention with volunteers is well explained

- The language used is clear

- The bibliographic references are adequate

- The strengths and weaknesses of the intervention are clearly explained, as    well as actions for improvement.

I look forward to your comments

Author Response

  1. I think it would be convenient to change the title of section: Literature review summary, for another title that better fits the content of the section.

The title was changed to ‘Literature summary as background to a new program’.

  1. Although in section 4.3 it is mentioned, it would be important to explain in greater depth the characteristics of the volunteers (training, age, etc.).

We added a broader description of the volunteers on page 14:

Their ages varied between 35 and 65. Twelve are Druze, six are Muslim, and two are Christian. Ten of them hold academic degrees (three are social workers and seven are teachers), eight have some form of higher education, and two are high school graduates.

  1. As the program is at an initial stage, it has not been possible to evaluate the long-term effects, but neither have short-term effects been observed, as was the initial objective? Two concrete examples are given in point 5.1, but it would be useful to present the results of the intervention in a more structured and concrete way.

We thank the reviewer for the important comment. As we intend to dedicate a separate manuscript to the program's effects, to make things clearer we deleted the paragraph on section 5.1 with the examples.
